# Genome-Wide Association Study on Imputed Genotypes of 180 Eurasian Soybean *Glycine max* Varieties for Oil and Protein Contents in Seeds

**DOI:** 10.3390/plants14020255

**Published:** 2025-01-17

**Authors:** Nadezhda A. Potapova, Irina V. Zorkoltseva, Alexander S. Zlobin, Andrey B. Shcherban, Anna V. Fedyaeva, Elena A. Salina, Gulnara R. Svishcheva, Tatiana I. Aksenovich, Yakov A. Tsepilov

**Affiliations:** 1Kurchatov Genomics Center, Institute of Cytology and Genetics SB RAS, Lavrentiev Av. 10, 630090 Novosibirsk, Russia; zlobin@bionet.nsc.ru (A.S.Z.); atos@bionet.nsc.ru (A.B.S.); sunday01@mail.ru (E.A.S.); gulsvi@bionet.nsc.ru (G.R.S.); tsepilov@bionet.nsc.ru (Y.A.T.); 2The Federal Research Center, Institute of Cytology and Genetics SB RAS, Lavrentiev Av. 10, 630090 Novosibirsk, Russia; fedyaevaav@bionet.nsc.ru (A.V.F.); aks@bionet.nsc.ru (T.I.A.); 3Institute of General Genetics RAS, Gubkin St. 3, 119333 Moscow, Russia

**Keywords:** soybean, *Glycine max*, oil, protein, GWAS, Russia, soybean breeding, phenotyping, genotyping, imputation

## Abstract

Soybean (*Glycine max*) is a leguminous plant with a broad range of applications, particularly in agriculture and food production, where its seed composition—especially oil and protein content—is highly valued. Improving these traits is a primary focus of soybean breeding programs. In this study, we conducted a genome-wide association study (GWAS) to identify genetic loci linked to oil and protein content in seeds, using imputed genotype data for 180 Eurasian soybean varieties and the novel “genotypic twins” approach. This dataset encompassed 87 Russian and European cultivars and 93 breeding lines from Western Siberia. We identified 11 novel loci significantly associated with oil and protein content in seeds (*p*-value < 1.5 × 10^−6^), including one locus on chromosome 11 linked to protein content and 10 loci associated with oil content (chromosomes 1, 5, 11, 16, 17, and 18). The protein-associated locus is located near a gene encoding a CBL-interacting protein kinase, which is involved in key biological processes, including stress response mechanisms such as drought and osmotic stress. The oil-associated loci were linked to genes with diverse functions, including lipid transport, nutrient reservoir activity, and stress responses, such as Sec14p-like phosphatidylinositol transfer proteins and Germin-like proteins. These findings suggest that the loci identified not only influence oil and protein content but may also contribute to plant resilience under environmental stress conditions. The data obtained from this study provide valuable genetic markers that can be used in breeding programs to optimize oil and protein content, particularly in varieties adapted to Russian climates, and contribute to the development of high-yielding, nutritionally enhanced soybean cultivars.

## 1. Introduction

Soybean is one of the most important leguminous crops in the world, and it is cultivated in numerous countries. The main soybean producers are concentrated in Asia (e.g., China and India) and South and North America (e.g., Brazil, AR, USA). Additionally, this crop is cultivated in Europe, including France, Germany, and Russia. Soybean is a major source of oil, protein, and other nutrients for human and livestock consumption. It is mostly used in the food industry to produce tofu, soy milk, and soybean oil. It might play an important role in the world’s food strategies in the future [1]. Other than that, soybean products are employed to produce clothes, plastics, fuels, and other things [2,3,4].

Due to the importance of soybean, a considerable number of genetic studies have been conducted with the objective to describe its population genetic characteristics [5,6,7], functional annotations and description of genes and their functions [8,9,10], genome-wide association studies (GWAS) of different soybean varieties and traits [11,12,13,14] and genomic selection for different traits [15,16,17].

A number of certain soybean traits are under special attention for selection. The protein and oil contents in seeds directly correlate with nutritional value, while weight and number of seeds per plant, ramification number, number of pods and grains per plant, plant height, and first pod insertion height are also important for harvesting soybeans using agricultural machinery.

The easiest way to understand the correspondence between genotypes and traits is to perform a GWAS that provides a set of statistically significantly associated candidate loci for further validation and selection studies. There are many studies that have discovered important agriculture loci for soybeans (*G. max*) from different regions [18,19,20]. These locus sets might differ and sometimes cannot be replicated on other datasets due to different origins, population structures, genetic structure of traits, etc. Therefore, the availability of GWAS results for soybean varieties from different regions is important for agriculture.

The protein and oil contents in soybean seeds have a significant negative correlation [21,22]. A number of studies have been conducted to identify the genetic loci associated with these traits. To illustrate, in the work [23], 15 novel loci associated with protein content and 24 loci associated with protein content have been identified using soybean datasets from different countries. The analysis of 877 soybean accessions grown in the USA showed three and five genomic regions associated with protein and oil contents, respectively [21]. Among the identified loci, there were those that had no effect on each other. This means that there is no negative correlation and provides an opportunity to use it in further studies leading to genomic selection. In [22], five loci were identified as being associated with protein content, and six loci were identified as being associated with oil content in seeds.

A significant number of studies have revealed that the major part of the associated loci is located on chromosomes 15 and 20 [21,22], regardless of the origin of soybean varieties under examination. The genes on these chromosomes that significantly affect protein and oil QTL have been described. A sugar transporter gene (*GmSWEET39*) was mapped to QTL on chromosome 15 [24]. CCT-domain gene, *POWR1* (Protein, Oil, Weight, Regulator 1), underlies a large effect on protein and oil QTL on chromosome 20. A candidate gene associated with protein content was identified by molecular and transgenic analyses. It was demonstrated that GmST05 (Seed Thickness 05), which is located on chromosome 5, not only positively regulated seed size but also influenced oil and protein contents.

No genetic analyses have been conducted on Russian varieties. We have previously performed a comprehensive analysis of the population structure and genetic diversity of the 175 soybean *G. max* accessions cultivated in Western Siberia and other regions of Russia [7]. It has been shown that the Russian varieties differ both from the known world collection and from each other. Consequently, this collection holds considerable promise for the identification of novel genes influencing protein and oil content in soybeans.

The majority of soybean GWAS studies were performed using data from SNP arrays. This means that only a relatively small set of SNPs compared to genome size has been involved in the studies. One potential solution to this problem is imputation, particularly given the availability of datasets with full soybean genomes [25,26]. To date, there is only one study that describes widely used tools (BEAGLE v.5.0, IMPUTE v.5, and AlphaPlantImpute) for genome imputation, and their results applied to soybeans [27]. Imputation provides the potential to find additional associated loci that may have been “hidden” in the array dataset.

In this study, we performed a genome-wide association study (GWAS) for two agriculturally significant traits, namely oil and protein content in seeds, using imputed genotypes of 180 Eurasian soybean varieties (*G. max*). In addition, we developed specific PCR-based markers for several key loci, including those near the *GmSWEET39* and *GmST05* genes that are associated with protein content. As these markers allow for the efficient screening of high-protein alleles, thus aiding breeders in marker-assisted selection, we set out to test the efficacy of these markers in the Russian population.

## 2. Materials and Methods

### 2.1. Soybean Population

We analyzed the dataset comprising 180 soybean (*G. max*) accessions. This dataset included 87 Russian and European cultivars and 93 breeding lines from Western Siberia. These accessions have been stored and multiplied in Western Siberia by the Siberian Federal Scientific Center of Agro-BioTechnologies of the Russian Academy of Sciences (SFSC RAS, Novosibirsk, Russia). Further information is provided by [7] (in Appendix A). In addition, 11 soybean cultivars were kindly provided by the Federal Scientific Center of Legumes and Groat Crops (FSC LGC, Orel, Russia). A list of soybean varieties studied with detailed information about each can be found in the article [28]. The majority of the analyzed accessions have been collected by breeders from the Siberian Federal Research Center of Agro-BioTechnologies of the Russian Academy of Sciences (SFSCA RAS) (Novosibirsk, Russia). These have been used as the main plant material for the selection of new cultivars. Novosibirsk’s location at 55 degrees north latitude makes it an atypical area for soybean cultivation. The conditions are extreme for a thermophilic and photoperiod-sensitive crop. Therefore, our study may also be of interest with regard to adaptation to atypical conditions.

### 2.2. Genotype Data

Genomic DNA was extracted from 3- to 4-day-old seedlings grown in Petri dishes using the CTAB method, as previously described in [29]. Genotyping of 180 soybean accessions was performed for 52,041 SNPs using the SoySNP50K iSelect BeadChip array [30] in the Genomic Centre of ICG SB RAS. The initial raw data were analyzed using the Genome Studio v.2 software (Illumina Inc., San Diego, CA, USA). Then, the results were converted to the PLINK format. The reference genome Wm82.a1 was downloaded from the SoyBase database [31] (https://www.soybase.org/tools/snp50k/, accessed on 30 September 2024) as this was the version on which the BeadChip array was developed. The raw file of 180 genotypes for 52,041 SNPs has been deposited in Zenodo [32].

### 2.3. Phenotype Data

In this study, we analyzed two phenotypes, oil and protein contents in seeds, measured for plants in 2021 and 2022 in two regions: Orel and Novosibirsk. The growing conditions of the plants have been previously described in [28].

The oil and protein contents of whole seeds were determined using an OmegAnalyzerG (Bruins Instruments, Weiler bei Bingen, Germany). This infrared analyzer also measured the moisture content of the seeds. All analyses were carried out in triplicate at ~14 g for each sample. In this study, the values of oil and protein contents are expressed as a percentage of dry weight.

In total, four datasets for each trait, comprising 180 samples each, were generated (Appendix A for oil and protein content, respectively). In each dataset, phenotypic values that were more than three times the interquartile range below the first quartile or above the third quartile [33] were excluded from the following analyses using R v.4.3.3.

### 2.4. Genotype Imputation

The imputation was performed in two steps. In the first step, our filtered data were imputed using the SoySNP50K genotypes of the USDA Soybean Germplasm Collection SoyBase database [34], which were downloaded from SoyBase (https://www.soybase.org/, accessed on 30 September 2024). For this imputation, we filtered data from the database with the following parameters: genotype quality per sample > 0.95, heterozygous rate > 0.05, minor allele frequency > 0.02, and genotype quality per SNP > 0.9. Moreover, we considered pairs of samples with identity-by-descent (IBD) values > 0.98 to be duplicated [35]. We also deleted all SNPs with inappropriate chromosome names. Subsequently, the resulting file was used for imputation using the BEAGLE v.5.4 software [36,37] with the following optimal parameters: effective population size (*ne*) = 10, the length of sliding window (*window*) = 200, the allele mismatch probability for the hidden Markov model (*err*) = 0.0005, and the minimum cM length of haplotype segments that will be incorporated in the HMM state space for a target haplotype (*imp-segment*) = 30. These parameters have been recommended for soybeans to ensure high imputation accuracy [27].

In the second step, imputed data were filtered (*r*^2^ > 0.7 and minor allele frequency > 0.1) and imputed using 302 resequenced samples [38]. We used the same optimal parameters as in the first step. The resulting SNPs were filtered by *r*^2^ > 0.7 and minor allele frequency > 0.01 and were used in further analysis.

### 2.5. Genome-Wide Association Analysis

A linear mixed model of trait inheritance, implemented in the GEMMA software [39] version 0.98.5 (25 August 2021), was used to analyze the associations between each phenotype and the SNP data. GWAS was performed for each region separately (Novosibirsk and Orel) and for both regions together. We used a similar methodology in our previous work, detailed in [40]. In essence, to analyze the trait measured at the four “region/year” points as a single combined phenotype, we quadrupled the initial sample size by including three duplicates for each original sample. These duplicates were considered as “genotypic twins” with different trait values. This approach makes it possible to have a larger sample size, which increases the statistical power of the study. In our model, we treated the “region” and “year” factors and their interaction as fixed-effects covariates. To our knowledge, this approach has not yet been used in other GWASs of soybean.

We conducted the analysis using a genomic relationship matrix reflecting the genetic similarity between each pair of lines. A genomic relationship matrix was calculated based on SNP genotypes with a minor allele frequency (MAF) > 0.05 in accordance with VanRaden’s method 2 [41] using the R *calc_gnrm*() function accessible via the following link (https://github.com/soloboan/GNRM_genomicpedigree/, accessed on 1 November 2024). A MAF-based filter of 0.01 was applied by GEMMA during the analyses. The Wald test was used to calculate *p*-values. Inflation factors were calculated as the median of the observed chi-squared test statistic divided by the expected median of the corresponding chi-squared distribution. The genome-wide significant threshold was calculated considering all genotyped SNPs before the imputation: *p*-value < 0.05/33171 = 1.5 × 10^−6^. The SNPs with *p*-value ≤ 1.5 × 10^−5^ were considered to have suggestive associations.

A clamping procedure to identify independent loci was performed using PLINK v.1.9.0-b.7.6. Clumping is used to keep a subset of SNPs that do not correlate with each other. We used the following parameters: --clump-p1 1.5 × 10^−5^ --clump-kb 500. The clumping procedure takes all SNPs that are significant at threshold p1 (index SNPs) and forms clumps of all other SNPs that are within a kb distance from the index SNP and that are in linkage disequilibrium with them, based on an r-squared threshold (default 0.50). Each SNP only appears in a single clump. This leaves only one representative SNP per LD region. For QQ and Manhattan plots, the R library qqman (v.0.1.8) was used.

### 2.6. Phenotypic and Genetic Correlations, Heritability Estimates

The heritability for each trait and the genetic correlation between them were calculated using the REML function from the GCTA software v1.94.1 Linux. Pearson’s phenotypic correlations were calculated between year-averaged traits for each region.

### 2.7. The In-Silico Interpretation of GWAS Results (Post-GWAS)

In post-GWAS, for each of the found significant loci (i.e., SNP), we searched for the nearest gene located near its coordinate using the SoyBase genome annotation platform (https://legacy.soybase.org/dlpages/#genomesequence, accessed on 1 November 2024). This was conducted in order to extract the gene IDs in the required format for subsequent analysis. Then, we searched for the functional annotation for each nearest gene located near each significant locus using the SoyBase Gene Annotation Tool (https://legacy.soybase.org/genomeannotation/, accessed on 1 November 2024). This tool gives comprehensive information in GO Annotation format about biological processes in which genes and their products are involved, molecular function, and cellular components where its products can be found. We also checked whether or not significant loci detected in this study had already been found using comprehensive GWAS datasets from SoyBase (https://data.soybase.org/Glycine/max/gwas/, accessed on 1 November 2024), the GWAS QTL tool (https://legacy.soybase.org/GWAS/list.php, accessed on 1 November 2024) and information from other large-sample GWAS papers not included in SoyBase [22,42,43,44].

### 2.8. Development of DNA Markers and Polymerase Chain Reaction

In the present study, we developed DNA primers for the *GmST05* gene (see Appendix A), which has previously been investigated in [45]. These primers flank the site of a 21 bp deletion within *GmST05* and give the PCR products of 139 and 118 bp in length for the alleles responsible for low- and high-protein contents, respectively (Appendix A). Furthermore, using dCAPS Finder 2.0 (http://helix.wustl.edu/dcaps/, accessed on 1 November 2024), we developed the dCAPS marker that allows distinguishing two different alleles of the *GmSWEET39* gene described in [24].

The PCR mixture, with a total volume of 25 μL, contained 10 mM Tris-HCl, pH 8.5, 50 mM KCl, 0.1% Tween 20, 2 mM MgCl_2_, 0.25 mM of each primer, 50–100 ng of DNA, and 1 U Taq DNA polymerase (BiolabMix, Novosibirsk, Russia). For *POWR1* gene, PCR was performed following the protocol described in [46]: 5 min at 95 °C; 35 cycles (95 °C, 15 s; 55 °C, 10 s; 72 °C, 40 s); and 1 min. at 72 °C. For the *GmST05* and *GmSWEET39* genes, PCR was conducted in accordance with the following protocol: 5 min at 95 °C; 12 cycles (95 °C, 15 s; 65 °C, 10 s; 72 °C, 10 s); 24 cycles (95 °C 15 s; 60 °C, 10 s; 72 °C, 10 s); and 1 min. at 72 °C.

For the dCAPS marker, a DNA restriction digestion was carried out in a 20 μL reaction mixture comprising 8 μL of PCR products, 2 μL of 10× restriction buffer, and 1 U Taq I restriction enzyme (SibEnzyme, Novosibirsk, Russia). The mixture was incubated for three hours at 65 °C.

The PCR and restriction products were separated in a 2% agarose gel with ethidium bromide. The results of electrophoresis were visualized and photographed in UV using the Gel Doc™ XR+ system (Bio-Rad Laboratories, Inc., Hercules, CA, USA).

## 3. Results

### 3.1. Characteristics of Studied Lines

The quality control of the genotype data was performed using the PLINK 1.9 software [47] (https://www.cog-genomics.org/plink/1.9/, accessed on 1 November 2024) and the following filters: genotype quality per sample > 0.95, genotype quality per SNP > 0.95, minor allele frequency > 0.01, and heterozygosity > 0.3. Consequently, 180 samples and 35,689 SNPs passed quality control.

The population structure and different population genetic characteristics of the studied dataset are comprehensively described in [7]. The genomic relationships among the soybean varieties were presented graphically as a dendrogram constructed using Ward’s hierarchical agglomerative clustering method (Appendix A). This illustrates the presence of four clusters that coincide with those previously described in our previous population structure analysis of these accessions [7].

### 3.2. Imputation

The imputation was performed in two steps. In the first step, our data were imputed using the reference data from the SoyBase database: 13,545 samples and 39,551 SNPs after quality control. As a result, after filtration, 33,171 SNPs were used for the second step. In the second step, using 302 resequenced samples, we imputed and obtained 3,997,265 SNPs after final filtration (*r*^2^ > 0.7 and MAF > 0.01).

### 3.3. Phenotypes

The distributions of phenotypic values characterizing protein and oil content in soybeans are presented in Appendix A. Appendix A shows the descriptive statistics for these traits. The phenotypic correlations between protein and oil contents were negative: −0.26 and −0.51 in the Novosibirsk and Orel regions, respectively. The genotypic correlation, calculated on a pooled phenotypic dataset, was −0.52. The SNP heritability estimate is high for both traits under study: 0.649 (standard error (SE) = 0.048) for protein content and 0.667 (SE = 0.046) for oil content.

### 3.4. GWAS

A GWAS was performed for each of the analyzed traits, both separately and jointly by regions. The GWAS results (Manhattan and QQ plots) for the pooled traits combining phenotypes from the Novosibirsk and Orel regions for the entire period of interest are presented in Figure 1. The GWAS results for each region are shown in Appendix A. Using the GWAS results, we calculated inflation factors, which yielded values of 0.942 and 0.984 for protein and oil contents, respectively.

Using the clumping procedure, we identified eight independent loci for protein content and 25 for oil content associated at the 1.5 × 10^−5^ threshold (Table 1). Eleven of them (one for protein content on chromosome 1 and ten for oil content on chromosomes 1, 5, 11, 16, 17, and 18) passed the genome-wide significance level.

### 3.5. Screening for the Presence of Genes Affecting Protein Content

The set of DNA markers for three genes (*POWR1*, *GmST05*, and *GmSWEET39*) was tested on the studied collection of 180 cultivars, including those of Siberian selection. The results showed that the *POWR1* allele, responsible for low-protein content, is present in all of the studied cultivars. The high-protein content allele was detected only in a single accession of wild soybean *G. soja*. The *GmST05* allele of increased protein content (Hap2) was found in four Russian cultivars: Gribskaya-12, Svetlaya, Belgorodskaya-48, and Annushka, along with several lines/varieties of foreign selection (No. 840-5-3, No./M 4888, ZS-8, Nordic, Morsoy). The PCR results for three markers of the set of varieties from the studied collection are presented in Appendix A. Thus, in the analyzed population of 180 soybean accessions, six accessions were found to carry the “high-protein” alleles of *GmSWEET39* and nine accessions—*GmST05*. Moreover, wild soybean *G. soja* was observed to carry “high-protein” alleles of the *GmSWEET39* and *POWR1* genes.

## 4. Discussion

In this study, we analyzed a collection comprising 180 accessions. We identified one locus as being significantly associated with protein content (*p*-value = 1.42 × 10^−6^), while 10 loci as being significantly associated with oil content (*p*-values ranging from 7.39 × 10^−9^ to 1.30 × 10^−6^). We analyzed all the loci significantly associated with oil content from both the Novosibirsk and Orel regions and found that these loci corresponded to seven different genes.

There is a significant environmental difference between the two studied regions. The climatic conditions of the Novosibirsk region, in contrast to the European part of Russia, are characterized by significant fluctuations in mean monthly and absolute temperatures, a longer daylight period in summer (e.g., in July, the day length in Novosibirsk is 30 min longer than in Orel), and an overall shorter summer. The average temperature in July is 18–20 °C, which is 5 °C lower than in Orel (https://yourastroguide.ru, accessed on 1 November 2024). Negative factors that may affect protein content include low night temperatures (13–15 °C) in the Novosibirsk region, which can drop even lower on certain days due to the sharply continental climate. The influence of these climatic factors on the protein and oil content of soybean seeds has been repeatedly demonstrated. It was shown that high oil content is observed under conditions of increased moisture and relatively low temperatures, while higher protein content is associated with dry weather and elevated temperatures [48,49,50]. We conducted GWAS analyses for each region, yet no region-specific loci were identified. We can speculate that all the genes identified increase the soybean’s resilience to various environmental stressors and harsh conditions.

Due to the incomplete state of soybean genomics and functional annotation, precise information regarding the role of each gene in cell and plant functioning remains limited. Nevertheless, we utilized an annotation containing Gene Ontology information in order to obtain a comprehensive picture of the genes nearest to each locus. The only significant locus associated with protein content (glyma.Wm82.gnm1.Gm11_30946582) is located near the gene encoding CBL-interacting protein kinase. This gene is involved in a number of various functions, including protein phosphorylation, signal transduction, protein kinase activity, serine/threonine kinase activity, ATP binding, and transferase activity related to phosphorus-containing groups. In soybeans, it has been shown to play an important role in response to environmental stresses, particularly drought, osmotic [51,52], and salt stresses [53]. One possible biological mechanism that may explain the effect of this gene on protein content is that it enhances soybean resistance to various environmental stresses and stimulates the accumulation of beneficial nutrients in the seed, thereby promoting the development of protein-rich soybean seeds.

Many previous studies have reported that major candidate genes associated with protein content are located on chromosomes 5, 15, and 20, and major QTLs are found on chromosomes 15 and 20 [21,22,45]. However, our GWAS analysis did not identify these loci in our population. We conducted an additional check for the presence of these previously described genes using markers developed earlier and in this study (see Appendix A). The GWAS results presented here are reliable because we used a specific linear mixed model of trait inheritance based on the powerful “genetic twins” approach to data integration. This approach allows the analysis of a combined trait that includes all “year/region” measurements, thereby increasing the power of the analysis by increasing the sample size. In addition, the GWAS analysis was performed on imputed genotypic data obtained using the BEAGLE v.5.4 software with optimal parameters. The high accuracy of imputation has been demonstrated in a soybean population in [27]. We suggest that the differences could be due to the population structure and the specificity of the varieties studied. We also studied a major locus on chromosome 15, identified as the *GmSWEET39* gene, which encodes the SWEET transporter [24]. For this gene, we investigated its casual variation in our collection. The low-protein allele is characterized by a deletion of the CC dinucleotide in the coding region, resulting in a truncated non-functional protein. This mutation decreases the total protein content by about 2% while simultaneously increasing the oil content by the same amount. Conversely, the high-protein allele reduces oil content, allowing selection to be directed toward either trait. The low-protein allele is predominant among American varieties [24] and, as our study revealed, in Russian varieties as well. However, we identified an alternative allele in several foreign varieties that have been adapted to Russian-like climatic conditions. Of the 180 accessions, only four (3.3%) carry the high-protein allele.

Recently, another candidate gene affecting protein content, namely *POWR1*, has been identified [46]. This gene encodes a CCT domain-containing protein and is located in a major QTL on soybean chromosome 20. The *POWR1* allele associated with low-protein content contains an insertion of a short transposon within the CCT domain, while the high-protein allele that lacks this insertion is the ancestral form commonly found in wild soybeans. A codominant marker for both alleles was developed, which we used to analyze the varieties in our collection (Appendix A). The high-protein allele was found only in wild soybeans.

In the work [45], the *GmST05* gene was studied, showing pleiotropic effects on seed size and protein content. A mutant haplotype (Hap2) associated with increased protein content was identified in several Chinese varieties, with a 21 bp deletion in the *GmST05* promoter. To screen for both the normal (Hap1) and mutant alleles of *GmST05*, we developed primers flanking the deletion site, creating a codominant marker (see Appendix A). Analysis of the soybean collection revealed that only nine accessions out of 180 (5%) carried the high-protein allele.

Thus, using a collection of 180 soybean accessions with low frequencies of previously described high-protein alleles in *GmST05*, *POWR1*, and *GmSWEET39*, we identified a new gene associated with protein content, located on chromosome 11.

Locus glyma.Wm82.gnm1.Gm01_53263067 (gene ID Glyma01g41880) is located near a Sec14p-like phosphatidylinositol transfer family protein, which exhibits transporter activity. This protein family is known to sense, bind, transport, and exchange lipophilic substances between membranes [54]. Locus glyma.Wm82.gnm1.Gm11_9120632 (gene ID Glyma11g12770) encodes a BURP domain-containing protein, previously suggested to play a role in plant stress response [55].

With regard to the loci, glyma.Wm82.gnm1.Gm16_652511 to glyma.Wm82.gnm1.Gm16_656481, the nearest gene (gene ID Glyma16g00980), was identified as Germin-like protein 1. This class of glycoproteins is known for its involvement in various biological processes, including stomatal complex morphogenesis, cellular cation homeostasis, divalent metal ion transport, manganese ion binding, and nutrient reservoir activity [56]. These proteins are predominantly located in the extracellular matrix, extracellular region, and apoplasts.

For loci on chromosome 16 (genomic region 656777–666228), the nearest gene was Germin-like protein 9 (gene ID Glyma16g00990), which has functions such as cold response, stomatal morphogenesis, photosynthesis, cellular cation homeostasis, defense response to bacteria, and metal ion transport. This protein is located in the extracellular region, cell wall, nucleus, and apoplast.

For loci glyma.Wm82.gnm1.Gm16_678372 to glyma.Wm82.gnm1.Gm16_678446, the nearest gene (gene ID Glyma16g01030), encodes a Baculoviral IAP repeat-containing protein involved in zinc ion binding and acid-amino acid ligase activity. This protein functions primarily in the mitochondria and chloroplast envelope.

Locus glyma.Wm82.gnm1.Gm17_7710790 is located near a BHLH transcription factor (gene ID Glyma17g10290) involved in cell cycle regulation and development, affecting processes such as transcription, fatty acid catabolism, jasmonic acid metabolism, pollen tube growth, and seed dormancy.

Locus glyma.Wm82.gnm1.Gm18_57755304 is located near an ankyrin repeat-containing protein (gene ID Glyma18g48310) involved in iron ion transport, zinc ion response, and tolerance to salt and drought [57].

In this study, we employed imputed soybean genotypes, a novel and effective approach in soybean research that significantly increases the number of loci used for GWAS. Traditionally, SNP arrays were used, but they are limited to a few tens of thousands of SNPs. Our approach dramatically expands the number of loci, increasing the power of the GWAS analysis. This method can be applied to other soybean studies because the reference genotypes for imputation are freely available, and the imputation algorithm is easy to use. The “genetic twins” method, based on quadrupling the sample size, represents an efficacious approach to integrating data from disparate regions and years, thereby enhancing the study’s power and facilitating the discovery of novel, previously unidentified loci. It should be noted that imputation techniques should be used with caution, as they are based on the assumption of identical haplotypes between the reference sample and the studied cohort. It is recommended to use the sequenced reference data obtained from the same studied population to ensure genetic homology. The results of imputed SNP association should be verified by genotyping methods.

Overall, in this study, we analyzed the collection of Eurasian soybean accessions and revealed one locus significantly associated with protein content and ten loci associated with oil content. Furthermore, in the studied population, we verified three genes that have previously been described as associated with protein content using PCR. The data obtained are valuable for future soybean breeding programs focused on optimizing both oil and protein levels. It provides key genetic markers that can be used to enhance desirable traits in soybean varieties adapted to specific climates and conditions.

## Figures and Tables

**Figure 1 plants-14-00255-f001:**
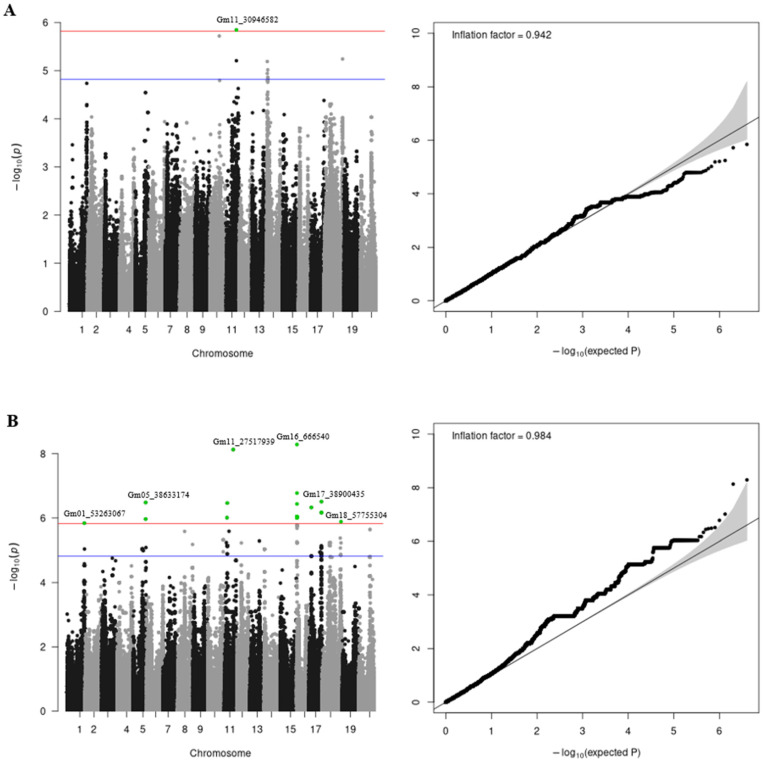
Manhattan and QQ plots for protein (**A**) and oil (**B**) content GWAS results for the pooled traits combining phenotypes from the Novosibirsk and Orel regions. The plots represent the −log10 transformed *p*-values. The red line indicates the significance level at *p* = 1.5 × 10^−6^. The blue line indicates the significance level at *p* = 1.5 × 10^−5^. Significant SNPs are shown in green. The top significant SNP per chromosome is annotated.

**Table 1 plants-14-00255-t001:** GWAS results for protein and oil contents at the 1.5 × 10^−5^ threshold.

Trait	Chr	SNP	A1	A0	AF	Beta	SE	P*
Protein	10	glyma.Wm82.gnm1.Gm10_28857346	T	A	0.230	−0.57	0.118	1.91 × 10^−6^
Protein	11	glyma.Wm82.gnm1.Gm11_30946582	G	A	0.235	0.59	0.121	**1.42 × 10^−6^**
Protein	14	glyma.Wm82.gnm1.Gm14_3287898	T	C	0.131	−0.56	0.124	6.44 × 10^−6^
Protein	14	glyma.Wm82.gnm1.Gm14_3289887	T	A	0.116	−0.55	0.125	1.14 × 10^−5^
Protein	14	glyma.Wm82.gnm1.Gm14_4736491	C	T	0.092	−0.66	0.148	9.58 × 10^−6^
Protein	14	glyma.Wm82.gnm1.Gm14_4745240	T	C	0.086	−0.61	0.140	1.46 × 10^−5^
Protein	18	glyma.Wm82.gnm1.Gm18_56804345	A	G	0.011	−1.50	0.328	5.71 × 10^−6^
Oil	1	glyma.Wm82.gnm1.Gm01_53263067	G	C	0.469	−0.39	0.081	**1.44 × 10^−6^**
Oil	5	glyma.Wm82.gnm1.Gm05_28366927	A	G	0.050	−0.64	0.142	8.87 × 10^−6^
Oil	5	glyma.Wm82.gnm1.Gm05_31043361	T	A	0.033	−0.91	0.205	1.02 × 10^−5^
Oil	5	glyma.Wm82.gnm1.Gm05_38633174	C	T	0.050	−0.70	0.135	**3.29 × 10^−7^**
Oil	5	glyma.Wm82.gnm1.Gm05_39499099	A	G	0.067	−0.46	0.103	8.19 × 10^−6^
Oil	8	glyma.Wm82.gnm1.Gm08_46435715	G	T	0.105	−0.51	0.113	6.55 × 10^−6^
Oil	10	glyma.Wm82.gnm1.Gm10_47564009	A	G	0.317	−0.38	0.081	4.74 × 10^−6^
Oil	11	glyma.Wm82.gnm1.Gm11_8270913	A	G	0.035	−0.91	0.183	**9.78 × 10^−7^**
Oil	11	glyma.Wm82.gnm1.Gm11_9120632	C	A	0.057	−0.72	0.140	**3.42 × 10^−7^**
Oil	11	glyma.Wm82.gnm1.Gm11_27517939	A	G	0.016	−2.17	0.371	**7.39 × 10^−9^**
Oil	12	glyma.Wm82.gnm1.Gm12_13640091	A	G	0.057	−0.65	0.143	5.88 × 10^−6^
Oil	14	glyma.Wm82.gnm1.Gm14_679279	C	G	0.020	−1.05	0.234	9.21 × 10^−6^
Oil	16	glyma.Wm82.gnm1.Gm16_666540	T	C	0.059	−0.66	0.111	**5.13 × 10^−9^**
Oil	16	glyma.Wm82.gnm1.Gm16_678372	A	C	0.050	−0.68	0.137	**9.93 × 10^−7^**
Oil	16	glyma.Wm82.gnm1.Gm16_708777	A	G	0.085	−0.56	0.110	**3.65 × 10^−7^**
Oil	17	glyma.Wm82.gnm1.Gm17_33884325	A	T	0.014	−1.05	0.237	1.14 × 10^−5^
Oil	17	glyma.Wm82.gnm1.Gm17_38648249	G	A	0.059	−0.52	0.116	7.19 × 10^−6^
Oil	17	glyma.Wm82.gnm1.Gm17_38677801	C	G	0.056	−0.50	0.112	8.25 × 10^−6^
Oil	17	glyma.Wm82.gnm1.Gm17_38900435	C	A	0.055	−0.58	0.112	**3.10 × 10^−7^**
Oil	18	glyma.Wm82.gnm1.Gm18_22353675	A	G	0.011	−1.26	0.279	7.03 × 10^−6^
Oil	18	glyma.Wm82.gnm1.Gm18_22813241	T	C	0.014	−1.21	0.266	5.84 × 10^−6^
Oil	18	glyma.Wm82.gnm1.Gm18_56821491	G	A	0.139	−0.35	0.080	1.33 × 10^−5^
Oil	18	glyma.Wm82.gnm1.Gm18_56862027	A	G	0.070	−0.56	0.129	1.40 × 10^−5^
Oil	18	glyma.Wm82.gnm1.Gm18_57755304	G	A	0.070	−0.53	0.108	**1.30 × 10^−6^**
Oil	20	glyma.Wm82.gnm1.Gm20_34390124	G	A	0.011	−1.14	0.239	2.28 × 10^−6^

The columns are: Chr—chromosome number, SNP—snp id, A1—minor allele, A0—major allele, AF—minor allele frequency, Beta—beta estimate, SE—standard error for beta, and P—*p*-value from the Wald test. P* *p*-values at the genome-wide significance level are shown in bold.

## Data Availability

The original contributions presented in the study are included in the article/Appendix A, further inquiries can be directed to the corresponding author.

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
