# Peer review of "Genome-Wide Association Study on Imputed Genotypes of 180 Eurasian Soybean Glycine max Varieties for Oil and Protein Contents in Seeds"

_plants, 2025, doi:10.3390/plants14020255_

Round 1
Reviewer 1 Report
Comments and Suggestions for Authors
The manuscript “Genome-wide association study on imputed genotypes of 180 Eurasian soybean Glycine max varieties for oil and protein contents in seeds” submitted by Potapova et al., was carefully reviewed. Glycine max (soybean) is a crop legume that globally constitutes one of the most important sources of animal feed protein and cooking oil. Although the authors identified 11 loci significantly associated with protein content and 10 loci associated with oil content, the values in the QQ chart are abnormal, the authenticity of the obtained loci lacks further verification. In addition, previous studies also identified key traits using whole genome re-sequencing data analysis of 106 soybean genotypes, which were not cited either. Based on above information, I cannot determine the originality of the author's manuscript.
Reviewer 2 Report
Comments and Suggestions for Authors
The manuscript entitled “Genome-Wide Association Study on Imputed Genotypes of 180 Eurasian Soybean Glycine max Varieties for Oil and Protein Contents in Seeds” presents valuable insights into genetic loci associated with oil and protein content in soybean seeds. This research is significant for advancing breeding programs aimed at optimizing nutritional quality and yield in Eurasian soybean varieties, especially under diverse environmental conditions. However, the following comments should be addressed.
Comments
- The Materials and Methods section, especially the genotype imputation and genome-wide association analysis subsections, could benefit from more details on the specific parameters used, as well as justification for their selection.
- The chosen p-value thresholds (e.g., 1.5 × 10^−5 and 1.5 × 10^−6) for significance should be explained in the context of this study to clarify their appropriateness.
- While the genes near significant loci are briefly mentioned, an expanded discussion on how these genes might influence protein and oil content could provide more insight, connecting the findings to potential biological mechanisms.
- A more detailed exploration of the genetic diversity within the soybean population studied, possibly with reference to a principal component analysis, would improve the context of the GWAS findings and their relevance across different environments.
- Consider adding a discussion on how environmental differences between the Orel and Novosibirsk regions might influence oil and protein content in soybean seeds. Environmental influence on gene expression is a key factor in breeding resilient crops.
- The manuscript could be strengthened by a deeper comparison with previous GWAS studies on soybean oil and protein content. This would help contextualize the novelty and importance of the loci identified in this study.
- Line 37: Consider adding details on the environmental conditions or agricultural significance of the soybean varieties to provide additional context.
- Line 96: Clarify why the threshold p-values were chosen for significant associations. Readers may benefit from knowing if this was based on prior studies or a specific statistical reasoning.
- Line 183: The annotation process for gene IDs could be clarified. Indicating the criteria used to select these annotations would provide better clarity.
- Line 211: In the Results section, it may help to expand on the rationale behind using the four-cluster structure in the population analysis to clarify the genetic diversity present in the studied lines.
- Line 245: In Figure 1, add clearer markers for significance levels to improve interpretation of the Manhattan and QQ plots.
- Clarifying some of the more technical terms, such as "clamping procedure" and "synthetic samples," would help readers unfamiliar with these terms to better understand the methods and results.
- Adding clear labels and potentially enlarging figures in the supplementary materials could improve readability. Including gene loci annotations directly on the Manhattan plots may also help readers quickly locate significant associations.
- The Conclusion section could be expanded to include practical implications of these findings, highlighting how breeders might utilize the identified markers in practical field settings.
- Including a brief paragraph on the limitations of using imputed data and the possible impact of this on the study’s results would provide transparency and help temper the interpretations.
- Ensure that all supplementary figures and tables referenced in the text are easy to locate and adequately described in the manuscript.
Reviewer 3 Report
Comments and Suggestions for Authors
The manuscript reports a GWAS study of 180 Russian soybean varieties using genome imputation tools, to search for loci potentially associated with two important traits-oil and protein contents in seeds. Respectively 10 and 1 loci have been identified for oil content and protein content traits. These loci are somehow linked to stress-response genes and may be implications of variety adaptation to extreme climate in Russia. This work is interesting and contains new findings differed from known soybean varieties. The manuscript is informative and well organized.
Reviewer 4 Report
Comments and Suggestions for Authors
Dear authors,
The manuscript by Potapova et al documents GWAS on imputed genotypes of 180 Eurasian soybean Glycine max varieties for oil and protein contents in seeds. The authors identified 11 novel loci significantly associated with oil and protein content in seeds (p-value < 1.5×10-6). Through further analysis, they found that protein-associated locus is located near a gene encoding a CBL-interacting protein kinase and the oil-associated loci were linked to genes with diverse functions. The authors concluded that the loci identified not only influence oil and protein content but may also contribute to plant resilience under environmental stress conditions. Moreover, they developed valuable genetic markers for improvement of oil and protein content in soybean cultivars. The writing is good. The figure and table are useful and help guide the reader as they learn about GWAS on oil and protein in soybean. I only have one comment as follows.
Minor comment
1. Fig.1 enlarge some letters in the left of panel A and B.
Author Response
Comment 1: 1. Fig.1 enlarge some letters in the left of panel A and B.
Response: FIxed, thank you!
Round 2
Reviewer 1 Report
Comments and Suggestions for Authors
Although the author has responded and made revisions, I still adhere to my previous viewpoint.
For QQ plot, in Figure 1A, most of the points are located below the diagonal, indicating that the observed P-values of most of the loci are lower than the expected values. The reason may be: (1) the model is unreasonable, and the P value is overcorrected, resulting in a low significance of the P value; (2) There is a linkage imbalance among a large number of SNP loci in the population, and the number of effective loci (loci without linkage imbalance between each other) is significantly lower than the actual number of loci, so the expected value of P value is underestimated;
In Figure 1B, most of the points are located above the diagonal, but the starting position is too early, which obviously does not conform to biological logic. It is speculated that the analysis model is unreasonable, the false positives of the data are too large, and the significance of the P-value observation value is overestimated.
Author Response
We would like to thank the Reviewer for the thoughtful feedback on the QQ plots in Figure 1A and 1B. We appreciate the detailed observations on the positioning of the points relative to the diagonal and the potential implications for the model and population structure.
We would like to clarify that the genomic inflation factor for both analyses falls within the acceptable range of 0.94-0.99, indicating that the model does not exhibit inflation or significant deflation of the test statistics. This suggests that systematic biases such as population stratification or overcorrection are unlikely to be driving the observed patterns. And it doesn't indicate that the model is unreasonable or that the proportion of false positives is large.
We analyse the imputed genotypes with a large number of SNPs (>4M) and most of the SNPs are on the diagonal line before -log10(p-value)<=1.5. Unusual behaviour of the tails of the distributions may indicate complex linkage disequilibrium patterns between SNPs.
This type of behaviour is common in other soybean GWAS, for example: doi: 10.1186/s12864-023-09687-6, doi: 10.3390/ijms20123041, doi: 10.3390/agronomy12020250, doi: 10.1111/nph.18792. Subjectively, the Q-Q plots in these examples look even worse. In light of the above, we disagree with the Reviewer that the Q-Q plots indicate an inadequacy of the model used.